# DEFINING DECEPTION IN DECISION MAKING

## ABSTRACT

With the growing capabilities of machine learning systems, particularly those that interact with humans, there is an increased risk of systems that can easily deceive and manipulate people. Preventing unintended behaviors therefore represents an important challenge for creating aligned AI systems. To approach this challenge in a principled way, we first need to define deception formally. In this work, we present a concrete definition of deception under the formalism of rational decision making in partially observed Markov decision processes. Specifically, we propose a general regret theory of deception under which the degree of deception can be quantified in terms of the actor's beliefs, actions, and utility. To evaluate our definition, we study the degree to which our definition aligns with human judgments about deception. We hope that our work will constitute a step toward both systems that aim to avoid deception, and detection mechanisms to identify deceptive agents.

## 1 INTRODUCTION

The growth in the capabilities of machine learning systems, particularly systems that directly communicate or interact with humans such as language models (Brown et al., 2020; Chowdhery et al., 2022; Wei et al., 2022), dialogue systems (Lewis et al., 2017a; He et al., 2018b; Wang et al., 2019; Kim et al., 2022), and recommendation systems (Liu et al., 2010; Kang et al., 2019), has led to increasing concern that such systems could be used to deceive and manipulate people on a large scale (Tamkin et al., 2021; Lin et al., 2021; Goldstein et al., 2023). For example, a language model could be trained to produce statements that elicit desired responses and then deployed through social media to influence a large number of people. This could be done in well-meaning contexts (e.g., public service announcements or education) or maliciously (e.g., deceptive marketing or social influence campaigns with political goals). These influences may not even be verbal: generative models could generate images that influence people in various ways.

Not all such influence is undesirable, and one might argue that very little social interaction is possible if no influence at all is allowed to take place. Therefore, a major challenge is defining the degree to which influence is intentional, aligned, and ethical. A basic requirement for such systems is to be non-deceptive toward the users that they interact with. Deception has been defined in multiple disciplines, including philosophy (Masip et al., 2004; Carson, 2010; Sakama et al., 2014), psychology (Kalbfleisch & Docan-Morgan, 2019), and learning theory (Ward, 2022), with prior machine learning work primarily focusing on supervised learning methods for *detecting* deception, as validated by human labels or judgement (Shahriar et al., 2021; Zee et al., 2022; Tomas et al., 2022). However, this perspective can be limiting when attempting to define deception in more complex settings where deception can be determined based on the effect you have on another agent. Additionally, trying to train agents to be less deceptive may require a decision-theoretic objective. While existing work mainly defines deception as the act of making false statements Shahriar et al. (2021); Zee et al. (2022); Tomas et al. (2022), the reality is that: (1) omissions can be inevitable because detailing the entire truth may be infeasible; (2) technically true statements can convey a misleading impression; (3) the listener might have prior beliefs such that a technically false statement brings their understanding *closer* to truth; and (4) statements that are technically *further* from the truth may lead the listener to perform *actions* more closely aligned with their goals. Hence, a complete definition of deception should go beyond simply considering the logical truth of individual statements. This complexity motivates introducing a definition of deception in the context of sequential decision making problems, where we can account for the listener's beliefs, belief updates, actions, and utilities. This definition is critical for classifying system behavior as deceptive, providing explicit objectives that minimize deception, and developing defense mechanisms in which users could use analysis tools that automatically detect potential deception.

We work toward this goal by proposing a concrete definition of deception in the framework of sequential decision making. In particular, we define this concept mathematically within a partially observable Markov decision process (POMDP) (Kaelbling et al., 1998) which models a potentially deceptive interaction between a speaker and a listener agent, and in which the speaker is the main agent, while the listener is folded into the environment dynamics. We show how the actions of the speaker, the changing beliefs of the listener, and rewards obtained by the listener can provide a way to measure deception. Specifically, our formalism models deception by examining how a speaker's communication indirectly influences a listener's downstream reward. In our model, this influence is mediated by the listener's beliefs, which are shaped by the communication and drive the listener's actions. We then test our general definition of deception with specific examples to illustrate how it can reflect human intuitions about deception when provided with an appropriate reward function for the listener.

In our experiments, we examine how deception is perceived in three interactions: a house bargaining interaction between a buyer and a seller, a consultation between a nutritionist and a patient, and small talk between two colleagues. Firstly, we conduct a user study in which participants rank simulated interactions along several axes of deceptiveness. Using these human labels, we learn a classifier that can flag a speaker as deceptive given the regret. We compare deception ratings between humans, our formalism, and LLMs to discern whether our definitions align with human intuition. Secondly, we build a dialogue management system and conduct a user study in which humans interact with the system and rank how deceptive they found these agents. Finally, to understand if we can quantify deception occurring in AI systems, we generate dialogues for a sample negotiation task (Lewis et al., 2017a) with an LLM and compare deception ratings between humans and our methodology.

Our contribution lies in defining deception in terms of different forms of regret, which measure the impact of a speaker's actions on a listener's downstream reward. These different regret metrics are obtains by defining the listener's reward function in different ways. This allows us to measure the "degree of deceptiveness" of an interaction between a speaker and a listener. Additionally, we show that our formalism can identify deceptive behaviors present in a given interaction executed by our dialogue management system. We hope that our work will represent a step toward both systems that aim to avoid deception, and detection mechanisms to identify deceptive agents.

## 2 DEFINING DECEPTION

Consider the potential for deception in the interaction in Figure 1: Luca expresses interest in buying a house that Sam is selling, leaking information about certain features they are most interested in, such as the number of bedrooms/bathrooms and the size of the rooms. Based on this, Sam can choose which facts about the house to share with Luca. Based on his resulting beliefs, Luca will decide whether to sign up for a house showing. In this way, Sam's utterance will result in a specific expected reward for Luca. Since Sam wants to entice Luca to sign up for a house showing, Sam can choose to explicitly lie about the house or omit undesirable information about the house (e.g., damages, noisy neighbors, or limited parking). More subtly, Sam can provide information that is technically true but misleading due to Luca's implicit beliefs, such as truthfully stating that the house has many bathrooms to create the impression that it is large (when it isn't). In many cases, it may be unclear whether Sam's action should count as deceptive and to what degree. To analyze potentially deceptive interactions such as this, we introduce our formalism in the following subsections.

### 2.1 PRELIMINARIES

We study deception in the context of an interaction between a speaker and a listener, which we represent as a partially observable Markov decision process (POMDP) Kaelbling et al. (1998). POMDPs are described by a tuple $\mathcal{M}^{\text{po}} = \langle \mathcal{S}, \mathcal{A}, \mathcal{T}, \mathcal{R}, \Omega, \mathcal{O}, \gamma \rangle$, where $\mathcal{S}$ is the state space, $\mathcal{A}$ is the action space, $\mathcal{T}$ is the state transition function, $\mathcal{R}$ is the reward function, $\Omega$ is the observation space, $\mathcal{O}$ is the observation function, and $\gamma \in [0, 1)$ is the discount factor. An agent executes an action $a_t$ according to its stochastic policy $a_t \sim \pi(a_t|b_t)$, where $b_t \in \mathcal{B}$ denotes the belief state based on the observation history up to the current timestep. Each observation $o_t \in \Omega$ is generated according to $o_t \sim \mathcal{O}(s_t)$. An action $a_t$ induces a transition from the current state $s_t \in \mathcal{S}$ to the next state $s_{t+1} \in \mathcal{S}$ with probability $\mathcal{T}(s_{t+1}|s_t, a_t)$, and an agent obtains a reward $r_t \sim \mathcal{R}(s_t, a_t)$. An agent's goal is to maximize its expected discounted return $\mathbb{E}\left[\sum_t \gamma^t r_t | s_0, a_0\right]$.

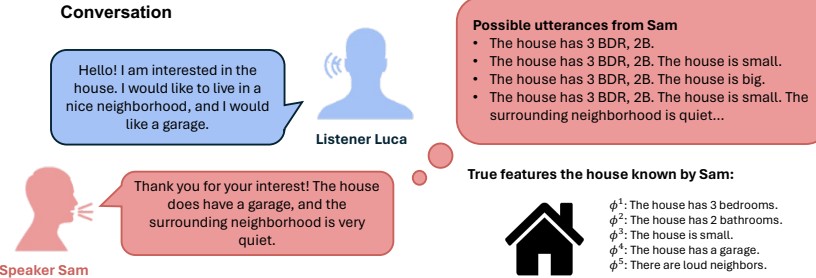

Figure 1: Sam is marketing a house to Luca. Luca's utterance shows they are concerned with the $\phi^4$ and $\phi^5$ features of the house. In response, Sam can choose an action $a_S$ from all possible combinations sharing or not sharing (lying or omitting) information. Finally, Luca selects an action $a_L$ (whether to go to a house showing), leading to Luca's agent-specific utility (corresponding to whether they will be happy they went). Depending on the $a_S$ action and its effect on the downstream utilities and beliefs of Luca, we can determine Sam's degree of deceptiveness.

## 2.2 THE COMMUNICATION POMDP

Consider an interaction between a speaker agent $S$ and a listener agent $L$, in which $S$ can perform actions that are potentially deceptive to $L$. The interaction between $S$ and $L$ proceeds as follows. $S$ observes the state of the world $s$ and sends a message $a_S$ to $L$. $L$ observes the message $a_S$ and updates their prior belief $b_L^0$ over their state using the observation $a_S$ and their model of the speaker's policy $\hat{\pi}_S$, which may not necessarily be the true speaker model (e.g. they may believe the speaker to be honest when they are not). Finally, they perform the action corresponding to the highest reward under their belief. We can formalize $L$'s behavior used in the transition dynamics of the communication POMDP as follows.

**Definition 2.1.** Given a model $\hat{\pi}_S(a_S|s_L)$ that $L$ has for the speaker $S$, the **listener model** is represented by the tuple $\langle \mathcal{S}, \mathcal{A}_L, \hat{r}_L, \Omega_L, b_L^0, b_L^{t+1} \rangle$:

- $\mathcal{S}$ is the set of world states over which $L$ maintains a belief $b_L$.
- $\mathcal{A}_L$ is the set of actions available to $L$.
- $\hat{r}_L(s_L, a_L)$ represents the listener's reward function (payoff) for performing action $a_L$ in state $s_L$. We explore choices of this reward function in Section 2.3.
- $\Omega_L = \mathcal{A}_S$ is the set of observations which $L$ may encounter, where each observation $o_L$ is a potentially deceptive communication action $a_S$ performed by $S$.
- $b_L^0(s_L)$ is the initial belief that $L$ has over the state $s_L$.
- $b_L^{t+1}(s_L|b_L^t, o_L) \propto \hat{\pi}_S(a_S|s_L)b_L^t(s_L)$ is the belief update of $L$ that represents the successor belief $b_L^{t+1}(s_L)$ after making observation $o_L = a_S$ under belief $b_L^t(s_L)$, where $\hat{\pi}_S(a_S|s_L)$ is the model that $L$ has for the speaker $S$.
- $L$'s policy is unknown to speaker: $\pi_L(b_L) = \arg\max_{a_L} \mathbb{E}_{s_L \sim b_L}\left[r_L(s_L, a_L)\right]$.

We now define the communication POMDP, where $S$ optimizes for a reward function that may incentivize deceptive behavior. Generally, $S$ may not know the beliefs of $L$ or $L$'s model of the speaker $\hat{\pi}_S(a_S|s_L)$.

**Definition 2.2.** Given a model for $L$ above, we define the speaker $S$'s **communication POMDP** as $\langle \mathcal{S}_S, \mathcal{A}_S, \mathcal{T}_S, r_S, \Omega_S, \mathcal{O}_S, \gamma \rangle$:

- $\mathcal{S}_S = \mathcal{S} \times \mathcal{B}_L \times \hat{\Pi}_S$, where $\mathcal{S}$ is the set of world states, $\mathcal{B}_L$ is the belief about the world state maintained by $L$, and $\hat{\Pi}_S$ is the set of speaker policies that may be assumed by $L$.
- $\mathcal{A}_S$ is the set of actions available to $S$, which may affect the belief $b_L$ of $L$.
- $\mathcal{T}_S(s_S^t, a_S^t, s_S^{t+1})$ is the transition function that represents the probability of transitioning to state $s_S^{t+1}$ after performing action $a_S^t$ in state $s_S^t$, which will depend on $L$'s model of $S$, $\hat{\pi}_S(b_L)$, as this will determine how $L$ will respond to $S$ and the state transition.
- $r_S(s_S^t, a_S^t, s_S^{t+1})$ captures the immediate reward for the speaker of transitioning from state $s_S^t$ to $s_S^{t+1}$ when $S$ performs action $a_S^t$. Note that this will implicitly depend on $L$'s response to $a_S^t$.
- $\Omega_S = \mathcal{A}_L \times \mathcal{S}$ is the set of observations made by $S$, where each observation $o_S$ is an action $a_L$ performed by $L$.

- $\mathcal{O}_S(s_S^t) = 1$ iff $o_S^t$ consists of the current listener action $a_L^t = \pi_L(b_L^t)$ and current external state $s$. Otherwise, $\mathcal{O}_S(s_S^t) = 0$. This models the fact that the speaker has access to the world state.
- $\gamma \in [0, 1]$ is the discount factor.

In many real-world scenarios, one can assume that $L$ does not think they are being deceived Kenton et al. (2021), and hence $S$ might assume a naive listener model for $L$. Even when this model is incorrect, it might provide for reasonable inferences for $S$ about whether or not they are being deceptive with respect to a "reasonable but naïve" listener. With this assumption, the communication POMDP immediately reduces to a fully observable MDP, which we show in Appendix A.

Note that our formulation of the communication (PO)MDP considers a single step of interaction: the speaker takes a communication action, the listener updates their belief, and then takes an action to receive the corresponding reward. While we consider this single-step formulation for simplicity of exposition, it is straightforward to extend the formalism into a sequential setting. If the listener asks a follow-up question, this would influence the listener's belief update $b_L^{t+1}(s_L|b_L^t, o_L)$ at the next step – e.g., if the listener asked a question that the speaker did not respond to directly, the listener might infer the answer was not what they might like.

## 2.3 DECEPTION FORMALISM

Given an interaction between a speaker and a listener, how do we determine whether the speaker has been deceptive? There are several intuitive notions of deceptive behavior: for instance, one could ground deception by considering whether $S$ negatively affects $L$'s beliefs (i.e., making their beliefs less correct), or the outcomes of $L$'s actions (i.e., making $L$ obtain less task reward, potentially for $S$ to get a higher reward for themselves). While the effect of $S$'s action on the reward of $L$ and on the belief of $L$ seem distinct, we provide a general definition for deception that represents both.

Our definition of deception aims to capture the nuances of indirect deceptive behavior, handle situations where providing full information is infeasible due to communication constraints, and provide a formalism that can be combined with existing decision making and RL algorithms. We measure deception in terms of the *regret* incurred by the listener from receiving the speaker's communication. This regret can be defined as a function of the speaker's actions, their effect on the listener's belief, and the effect of these updated beliefs on the listener's reward, providing a formalism that can be used as a reward function for the listener (e.g., to avoid deception) or as a metric (e.g., to measure if deception has occurred). By casting different intuitive notions of deception (i.e. the two sample reward functions) under the same regret umbrella, we provide a mathematical formalism that supports future algorithm design. Furthermore, the choice of reward for the listener allows granularity in specifying which types of outcomes one cares most about, whether it's inducing correct beliefs over some or all of the variables, or other goals.

We propose to measure the *degree of deceptiveness* of an agent through the formalism of regret, where a larger regret would indicate a more deceptive agent:

$$Regret(s, \pi_L, \pi_S) = \sum_{t=0}^{T} \mathbb{E}_{a_S^t \sim \pi_S, a_L^t \sim \pi_L(b_L^t)}\big[r_L(s, a_L^t)\big] - \sum_{t=0}^{T} \mathbb{E}_{a_L^t \sim \pi_L(b_L^0)}\big[r_L(s, a_L^t)\big]. \quad (1)$$

Here, $r_L$ is the reward of the listener when starting in state $s \in \mathcal{S}$, if $L$ and $S$ act according to $\pi_L$ and $\pi_S$ respectively. Under this regret formulation, the speaker is deceptive if they take an action that reduces the listener's expected reward relative to what the listener would have received had they acted according to their prior beliefs. In other words, we say deception has occurred if it would have been better if the listener had not interacted with the speaker at all. Hence, the speaker can be classified as *deceptive* if this regret is positive, *altruistic* if it is negative, and *neutral* if the regret is zero.

While on the surface it might seem strange to equate deception with causing suboptimal rewards for the listener, we argue that this general framework allows us to capture many of the intricacies of deceptive interactions, including "white lies" and true but misleading statements, if the reward function $L$ is selected carefully. In the following subsections, we explore ways to define $r_L(s, a_S)$ to capture our intuition about what constitutes deceptive behavior. We will show how the "logical truth" definition in fact is subsumed by our more general definition for an appropriate choice of reward, but our definition can also capture more nuanced situations.

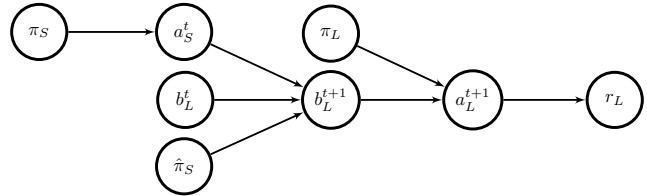

Figure 2: The interaction between the speaker and the listener is as follows: The listener $L$'s belief is updated based on $S$'s action (interpreted according to $L$'s model of $S$'s behavior $\hat{\pi}_S$). The listener will make a decision and receive reward based on their updated belief.

### 2.4 DEFINING UTILITIES FOR THE LISTENER

Depending on the scenario, a listener may place different value on obtaining accurate information and on making correct or generally beneficial decisions. In this section, we show how different intuitively reasonable notions of deception can emerge from our definition above, simply by making different choices for the listener's reward $r_L$.

The natural starting point for $L$'s reward is to make it equal to the "task reward" $\hat{r}_L$ (e.g., a house buyer might receive a higher reward for buying the right house). Defining the reward of $L$ in this way is reasonable in cases in which the "task reward" captures everything $L$ cares about. This could include utilities indicating that $L$ does not care about being deceived as long as it improves outcome.

**Deception as worsened outcomes:**

$$r_L(s, a_L) = \hat{r}_L(s, a_L), \tag{2}$$

where $\hat{r}_L$ is the listener's "task reward". The speaker is considered deceptive if the interaction with the listener leaves them worse off in terms of expected "task reward". The "task reward" captures the idea that people may care less about omissions or deception irrelevant to the task, such as Sam talking about how the house has a beautiful front porch when this is an embellishment and does not influence Luca's opinion of how valuable the house is to them.

However, we claim that the regret formulation is expressive enough to capture a variety of intuitive notions of deception. An obvious criticism might be that people might still feel deceived if they were "tricked" into making a good decision. However, this can be captured simply by redefining their reward: instead of receiving a reward only for a good decision, they also receive a reward for having an accurate belief over the state, or some subset of the state. For example, we use $r_L(s, a_L) = \hat{r}_L(s, a_L) + w b_L(s)$, where $\hat{r}_L(s, a_L)$ is the task reward and $w \in \mathbb{R}$ is a constant weight, the $b_L(s)$ term will provide for lower regret whenever the speaker changes the listener's beliefs to be more accurate, and higher regret when it makes their beliefs less accurate. Below we show how, for a specific choice of $r_L(s, a_S)$ in Equation (1), we can also capture the accuracy of beliefs in our metric for deception.

**Deception as leading to worse beliefs:**

$$r_L(s, a_L) = b_L(s), \tag{3}$$

where $b_L$ is the current listener belief, which we can obtain from the listener action as described in Appendix C.1. This definition can be thought of as a "score on a belief-accuracy test": consider an example scenario where $L$ is answering questions on an exam administered by $S$. As $L$'s expected value on this exam is the probability $S$ assigns to the correct answer, we can formulate $L$'s reward function as the proportion of questions they get correct on the exam. It is also straightforward to extend this construction to weight correct beliefs over some dimensions or even functions of the state more highly – for example, we might potentially define the listener's reward in the house example as the probability they assign to the true monetary value of the house, which is a derived quantity that depends on the house's features.

We've shown how $r_L(s, a_S)$ in Equation (1) can be defined for different notions of deception. By quantifying deception as regret, we can define deception based on the beliefs or downstream task reward of the listener which are induced by the speaker's actions. Additionally, we've shown how one could combine them in practice.

### 3 EXPERIMENTAL METHODOLOGY

The goal of our evaluation is to determine how well our proposed metric for deception aligns with human intuition. To that end, we have: **(1)** designed three scenarios to study deceptive behaviors;

| Scenario | Learned Regret (ours) | | | LLMs | | |
|---|---|---|---|---|---|---|
| | Task | Belief | Combined | GPT-4 | LLaMa | Google Bard |
| Housing Scenario | 0.34 | 0.67 | 0.70 | 0.19 | 0.11 | 0.02 |
| Nutrition Scenario | 0.17 | 0.25 | 0.37 | 0.16 | 0.01 | 0.01 |
| Friend Scenario | 0.26 | 0.37 | 0.48 | 0.19 | 0.07 | 0.11 |

Table 1: Summary of correlation values between human deceptive labels and learned task regret (ours), belief regret (ours), combined regret (ours), and deceptive labels three LLMs for three different real-life scenarios where deception might occur. A larger correlation value is indicative of a method that aligns strongly with human intuitive notions of deceptive behavior. We find that the housing situation has the least ambiguity when it comes to aligning with human notions of deception, with more ambiguity present for the nutrition and friend scenario. These results were statistically significant (p-value <0.001).

**(2)** developed an interactive dialogue management system where we can deploy agents that are deceptive to different degrees according to our proposed definition; **(3)** created a pipeline to measure the deceptiveness of responses from an LLM in a negotiation task.

For the first experiment, we ask humans to rate the deceptiveness of each interaction in a series of conversational scenarios, and provide comparisons by measuring the correlations between our approach as outlined in Equation (1), human ratings, and baseline evaluations by three state-of-the-art LLMs (OpenAI, 2023; Touvron et al., 2023; Google, 2023). For the second experiment, we evaluate our dialogue management system by conducting a user study to measure the correlation between human rating after interacting with the system and the deceptive regret of the policy deployed. For our third experiment, we use an LLM to generate negotiation dialogues based on a standard negotiation dataset (Lewis et al., 2017b), ask humans to label the deceptiveness in these negotiations and measure the correlation between human ratings and our deceptive regret. For our study with human participants, we received IRB approval and used CloudResearch Connect to recruit participants.

### 3.1 MEASURING DECEPTION IN CONVERSATIONAL SCENARIOS

We have designed three scenarios to capture how deception is perceived by humans in different contexts: a house bargaining interaction between a seller and a buyer, a consultation between a nutritionist and a patient, and small talk between two colleagues. These have been designed to consider different models of the listener, leading to differing ratings of deception (e.g., it is more deceptive to lie about features of a house than lie about your hobbies). Each scenario consists of three features that can be either true or false. A sample interaction is shown in Figure 4. We provide further details about the scenarios in Appendix D.

**Scenario generation.** We programmatically generate conversation scenarios for each situation described in Appendix D.1, consisting of listener preferences and speaker actions. Similarly to how prior work Bakhtin et al. (2022) translates symbolic moves into natural language for Diplomacy, we use an LLM (gpt-3.5-turbo) (Brown et al., 2020) to wrap "symbolic" POMDP communication actions from our model into natural text. We consider a setting in which the state consists of $k = 3$ features, with Luca "interested" in a random subset of these features. The features are considered independently by Luca, and there are no correlations between features.

**User study setup.** We show each of $N = 50$ users a series of 10 random scenarios for each situation (total of 1500 interactions), consisting of: 1) the true features (that are only known to Sam), 2) the prior belief $b_L^0$ Luca has about such features, 3) which features Sam revealed to Luca (given that the participants are aware of the true features, they can determine whether Sam was truthful or not), and 4) which features Luca cares about. For each scenario, participants were asked to rate whether they believe Sam's behavior is deceptive on a 1-5 Likert scale, from "Strongly Disagree" to "Strongly Agree". We describe our scenario sampling mechanism and provide details in Appendix D. Figure 4 provides examples of the interactions users will see for the three real-life scenarios.

### 3.2 DEVELOPING A DIALOGUE MANAGEMENT SYSTEM

To understand how a human's perception of deception changes upon interaction with a system, we have built a dialogue management system as shown in Figure 3 to simulate a real-world scenario where a human could be easily deceived. We chose to demonstrate this through the housing scenario detailed in Figure 1, where a human must input their preferences and engage in dialogue with an online representative (our model) who will share information about an available home. For this study, we have added further complexity to the scenario by increasing the number of features to eight and

including correlations between features, such that the human user cannot determine if the agent is lying within a few rounds. Similar to the previous setup, we use an LLM (gpt-3.5-turbo) (Brown et al., 2020) to wrap actions from our model into natural text, this time selecting actions that either maximize or minimize the deceptive regret (task and/or belief utility) at random based on the house preferences. For our user study to obtain deceptive human ratings, we have $N = 30$ users interact with our system.

### 3.3 Detecting deception in LLM-generated negotiation

To demonstrate how our definition can be used to quantify deception in dialogue, we use an LLM (gpt-3.5-turbo) to generate 30 negotiation conversations as motivated by successes in using LLMs to generate synthetic data (Bai et al., 2022; Park et al., 2023a; Abdulhai et al., 2023). We focus on the Deal or No Deal task Lewis et al. (2017b). Here, two agents, Agent 1 (speaker) & Agent 2 (listener), must split an inventory of three items between them. We have modified the original task setup such that Agent 1 is aware of the point values of Agent 2, but Agent 2 is not aware of the true point values of Agent 1. Instead, Agent 2 has a prior belief over Agent 1's point values, and Agent 1 can communicate in ways that are truthful or deceptive about their preferences. A deceptive agent might seek to gain an advantage by implying they value some items more or less than they really do. To encourage diversity in the conversations, we instantiate each agent with a different utility function according to which it negotiates. To compute the deceptive regret for the conversation, we use chain of thought prompting (Wei et al., 2023) to ask questions about the negotiation to determine the prior belief of the listener, the posterior belief of the listener at the end of the conversation, and the speaker's actions (i.e., shared point valuations). For our user study, we have $N = 30$ humans provide deceptive human ratings. A sample negotiation dialogue is shown in Figure 3, and we have provided further details of our setup in Appendix G.

### 3.4 Evaluation

We explain the results from our three experiments below.

**Q1: Does our definition of deception align with human judgment?** We compare human deception scores from our user study against regrets calculated as per Equation (2) and Equation (3) by computing their correlation as shown in Table 1. We combine two reward terms (labeled "Combined") to see whether that is able to better capture human intuitive notions of deception. To do so, we regress human deceptiveness labels on both our regret metrics individually and jointly. While using both reward terms in conjunction improves predictions, the majority of the predictive power comes from the belief regret $b_L(s)$. We largely find that a combined regret formulation better captures human intuitive notions of deception across all three scenarios, confirming our hypothesis from Section 2.3 that both belief and task reward contribute to improving the correlation with human judgment. For the housing scenario, we find a significant correlation of 0.67 between human responses and that shown by belief-based regret, and a correlation of 0.34 between human responses and task-reward-based regret. This matches our intuition that humans primarily focus on the truthfulness of statements more than just outcomes (which is closer to a purely utilitarian perspective). We find the least correlated values shown for the nutrition scenario, which might indicate that due to ambiguity in the listener's observation model, humans may be noisy when discerning whether deception is taking place. We found that for these two scenarios, humans ranked interactions as overall being less deceptive, whereas our model labeled them as being more deceptive comparatively. This might be indicative that there might be additional reward terms that may capture the conservative labeling of humans and the subjectivity of defining deception depending on the scenario.

For multi-step conversations occurring as part of the dialogue management system, we found the correlation between deceptive ratings from humans and our formalism to be 0.72 for belief utility and 0.45 for task utility respectively, slightly higher than the correlations of 0.67 and 0.34 when users observe interactions as shown in Table 1 for the housing interaction. This shows that our deception metric has the ability to scale when the conversation contains the complexity present in the real-world, including correlations in beliefs and

**Q2: How do LLM judgments compare at discerning deception?** LLMs have been shown to sometimes be successful in performing data annotation, sometimes even surpassing human annotator quality (Pan et al., 2023; He et al., 2023; Wang et al., 2021). We explore how well LLM evaluations correlate with human judgments about deceptiveness in Table 1. The purpose of this evaluation is to examine whether or not it is trivial to infer the degree of deception in these statements. In particular,

we use three state-of-the-art LLMs (OpenAI, 2023; Touvron et al., 2023; Google, 2023) with the same prompt that was given to the human annotators, asking whether each given interaction is deceptive – and compare the LLM deception labels with those in the user study. We find that even very large, state-of-the-art LLMs, such as GPT-4, do not make deceptiveness judgments on these examples that align as well with user intuition as even the worst choice of reward for our approach. Overall, we find GPT-4 aligning more than Google Bard and LLaMa across all three situations, respectively. Overall, these experiments validate our hypothesis that our formalism can be effective in estimating the "degree of deceptiveness" of human interactions and that our proposed formulation aligns with human intuition. For an initial exploration of how to create non-deceptive agents, see Appendix D.2.

**Q3: How can we leverage a regret theory of deception to measure deception from LLMs?**
Due to the increasing concern that LLMs could be used to deceive and manipulate people on a large scale, we generated sample negotiations for the Deal or No Deal Lewis et al. (2017a) to demonstrate a case of deception. Although we had humans only rate 30 dialogues, we generated a total of 500 dialogues to ensure a range of diverse strategies employed by agents in conversation, and by extension, a larger range of deceptive regret values. We have found there to be a correlation of 0.82 between human ratings of deception for the subset of conversations and our deceptive regret model, showing that human intuition agrees with the labels we assign. We expect that these labels may be leveraged as rewards for learning deceptive and non-deceptive LM models in the future.

## 4 RELATED WORK

**Deception in social psychology and philosophy.** Deception has been defined and analyzed through philosophy (Masip et al., 2004; Martin, 2009; Todd, 2013; Fallis, 2010; Mahon, 2016; Sakama et al., 2014) and psychology (Kalbfleisch & Docan-Morgan, 2019; Zuckerman et al., 1981; Whaley, 1982). To our knowledge, the most comprehensive definition (Masip et al., 2004) integrates the work of several researchers on lying (Coleman & Kay, 1981) and deceptive communication (Miller & Stiff, 1993), considering deception as the act of deliberately hiding, altering, or manipulating information—through words or actions—to mislead others and maintain a false belief. However, these definitions are mostly qualitative and are difficult to turn into precise mathematical statements that could be leveraged as objectives for training autonomous agents that embody various degrees of deception. Our definition formalizes deception within POMDPs, and is designed to be used as a reward function to build non-deceptive agents. Importantly, our work is inspired by work in moral psychology that contrasts utilitarianism, which aims to maximize the overall well-being (Driver, 2022), with deontological philosophies, which posit inviolable moral rules that do not vary with the situation (Greene, 2007). Our formalism allows both utilitarian and belief perspectives of deception to be represented by a regret formulation that can be used as a utility measure. Several works also define deception depending on whether or not the listener is aware (i.e., coercion and rational persuasion) (Todd, 2013) or unaware (i.e., lying or manipulation) (Noggle, 2022) of deceptive influence. Our work represents both as we do not make any assumptions about the listener (i.e., the listener uses a model that may or may not assume the speaker often lies).

**Deception in language models and mitigation.** With the development of LLMs with emergent capabilities (Wei et al., 2022), there has been a growing concern that these models may exhibit deceptive tendencies (Kenton et al., 2021). This occurs due to the model having misspecified objectives, leading to harmful content (Richmond, 2016) and manipulative language (Roff, 2020). Our work can potentially help address this misalignment Amodei et al. (2016) by providing a definition of deception that can modify the objective function or constrain the behavior of reinforcement learning agents to avoid deceptive tendencies. Several methods have focused on detecting deception in human text by using language models with manual feature annotation (Fitzpatrick & Bachenko, 2012), contextual information (Fornaciari et al., 2021), and textual data in a supervised manner (Shahriar et al., 2021; Zee et al., 2022; Tomas et al., 2022). These methods have been extended to detecting deception in spoken dialogue by learning multi-modal models through supervised learning (Hosomi et al., 2018; Soldner et al., 2019) and asking questions to improve estimates (Tsunomori et al., 2015). However, they may not cover the range of deceptive capabilities of LLMs as they only classify each utterance independently. Our work instead takes advantage of the sequential nature of interactions in AI systems in defining deception. We also differ from work on adversarial attacks Franzmeyer et al. (2023); Tondi et al. (2018) as we provide a general regret formulation under which the deceptive behavior of the speaker can be defined, quantified, and used as a way in which to label utterances in conversations with varying levels of deceptiveness. With respect to work on training agents to be

non-deceptive Hubinger et al. (2024), we would like to acknowledge that our formalism allows a system designer to capture the nuance in defining deception depending on the scenario.

**Deception in multi-agent systems and robotics.** Our work approaches deception from the view of sequential decision making problems, considering the effect of communication actions on a listener's beliefs. While expressing deception as changes in beliefs has been examined in prior work (Taylor & Whitehill, 1981; McWhirter, 2016; Gmytrasiewicz, 2020; Ward et al., 2023), our work converts belief-based definitions of deception into utility measures that can be used in reinforcement learning to avoid deceptive tendencies. Moreover, recent works Sarkadi et al. (2019); Adhikari & Gmytrasiewicz (2021); Ederer & Min (2022); Sarkadi (2018) have used communication or game theory to model deception of an agent with a theory of mind under uncertainty, and other game theoretic approaches Santos & Li (2009); Chelarescu (2021); Aitchison et al. (2021) have analyzed deception from a utilitarian perspective. Masters et al. (2021) has provided a qualitative account of deception in AI, and Park et al. (2023b) defines deception as the inducement of false beliefs when trying to achieve an outcome other than the true one. In contrast, our work provides a general framework that captures both belief-based and utility-based deception and quantifies deception as a continuous quantity, allowing us to measure the "degree of deceptiveness" of a speaker toward a listener. Additionally, while these methods assume that the speaker is intentionally deceptive by using a theory of mind, our work assumes that the speaker can be intentionally or non-intentionally deceptive, which depends on both the specific setting at hand and whether or not the speaker can access ground truth information. Lastly, several works have studied deception in non-verbal behavior, such as robot motion planning that deceives a person or makes it hard to infer intentions (Wagner & Arkin, 2011; Shim & Arkin, 2012; 2013; Dragan et al., 2015; Tomas et al., 2022; Ayub et al., 2021; Masters & Sardina, 2017). While our work approaches deception from the view of sequential decision making, it makes no assumptions on the action space, allowing it to be defined for both symbolic and textual forms of communication.

## 5 LIMITATIONS

We would like to acknowledge some limitations of our approach. Our formalism may inaccurately classify situations as deceptive when the speaker is simply suboptimal, leading to poor outcomes due to incompetence rather than intentional deceit. This misclassification occurs because our metrics might label such behavior as deceptive. If the speaker is modeled incorrectly, such as assuming they have complete knowledge when they do not, the resulting inferences about deceptiveness can be highly misleading. For example, a speaker might intend to deceive (attempting to lie and guide you towards a poor outcome) but accidentally convey the truth, leading to a better outcome. In such cases, the speaker would be wrongly classified as non-deceptive because their unintentional truthfulness resulted in a high reward. Moreover, our technique requires access to the ground truth state (and thus, a notion of what is true and false in the speaker's communication). We would like to note that many real-life situations assume a naieve listener who does not expect deception to occur, or that the speaker has full access to the state and can influence the listener in the way they intend. However despite this limitation, we believe that if we are not able to define deception under these simplifying assumptions, there is little hope to address more challenging settings with these assumptions relaxed. Lastly, we would like to acknowledge that we considered generalization to real-world scenarios when defining deceptive behavior. The scenarios we considered were designed to be simple enough to be quickly understood by humans, but complex enough to capture real-world behaviors. To consider more complicated scenarios, we generated dialogues for a well-known negotiation task, and our procedure could also be implemented for other similar benchmarks and datasets He et al. (2018a); Wang et al. (2020).

## 6 DISCUSSION

We cast deception from the lens of impacts on a listener's beliefs and resulting actions/task rewards. We found that a belief regret model, looking at the extent to which the listener more or less strongly believes in the correct state after interacting with the speaker, significantly correlates with users' subjective ratings of deception. Interestingly, the impact on the task reward of the resulting listener actions is a lot less predictive. Of course, this is just a start. Future research is needed to understand where the correlation breaks and what nuances explain what real people find deceptive. If the belief gets slightly worse, but the belief over aspects of the state that are actually relevant to the task reward gets better, is that still considered deceptive? This type of question presents a fruitful avenue for future investigation.

## 7 ETHICS STATEMENT.

We acknowledge that our formalisms may pose non-negligible ethical risks. They could be especially dangerous if used for targeted deceptive advertising, recommendation systems, and dialogue systems. We discourage the use of deceptive AI systems for malicious purposes or harmful manipulation. We hope this research provides grounding for how to define deception in decision making and build systems that can mitigate and defend against deceptive behaviors from both humans and AI systems. This work offers a concrete definition of deception under the formalism of decision-making. We expect our work to only be a step in the direction of formally quantifying and understanding deception in autonomous agents: while our definitions provide a working formalism, they may leave open edge cases. A key area of future work is to generalize these definitions to settings that reflect realistic domains of machine learning, such as dialogue systems, robotics, and advertising. Large-scale applications may include reward terms that prevent deception and detection methods. Exploring these applications may not only lead to practically useful systems aligned with human values but also suggest ways to formalize deception in autonomous agents.

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
