# OpenReview forum: "Defining Deception in Decision Making"
_ICLR.cc/2025/Conference — Submitted to ICLR 2025_

### Official Review · Reviewer_QS2c · 2024-10-25

**Soundness:** 4
**Presentation:** 2
**Contribution:** 2
**Rating:** 3
**Confidence:** 3

**Summary:**

The paper provides a formal definition of deception in a setting where a Speaker and a Listener interact with each other over multiple rounds. The goal of the proposed definition is to capture the Speaker's degree of deception by measuring to what extent its action(s) affect the Listener's beliefs and individual utility. The framework that is used to formalize the Speaker-Listener interactions is a POMDP variation, termed as Communication POMDP, where the Listener is modeled as part of the environment and the Speaker plays the role of the acting agent. Under this decision making framework, the Speaker's degree of deception is formalized through a function that measures the Listener's regret w.r.t. the accuracy of its beliefs and its personal utility, that results from the Speaker's actions. Finally, the paper includes extensive experimentation over a diverse set of scenarios that aims to showcase how well the proposed approach aligns with human intuition and how it compares to LLM baselines.

**Strengths:**

The paper is well-structured and the work well-motivated. Related work is sufficiently covered and the gap that the proposed definition aims to address is clearly highlighted.

The problem of detecting or preventing deception in AI systems is a very important one, and has become especially relevant these days with all the recent advancements in the field. The main idea promoted by this paper, i.e., that a formal definition for measuring deception is needed, is indeed crucial for making progress in this problem. To the best of my knowledge, this is the first work that attempts to provide such a definition that does not merely measure the truthfulness of agent's statements but instead looks deeper into the intricacies of the problem.

The Speaker-Listener setting on which this paper focuses does not of course capture all real-world decision making scenarios where detecting deception would be useful, but I find it general enough. The regret function proposed for measuring deception is simple to understand and quite reasonable, although far from complete as also mentioned in the Limitations section of the paper (see also Weaknesses below).

The biggest strength of the paper in my opinion is the experimental evaluation of the approach. I find it to be rigorous and very well-thought. Even though the proposed method does no do that well in all metrics, it significantly outperforms the LLM baselines.

**Weaknesses:**

**Presentation:** My main issue with this paper is Section 2.2 and the proposed POMDP framework. First, I found the presentation of this section to be quite poor: (a) Many things are not adequately explained, e.g., it is not explicitly mentioned that the world state does not change over time; (b) There are non-standard parts of this framework that are not revisited later on in the paper. This can hinder the understanding of the reader w.r.t. the role of these notions in the framework and why they are needed, e.g., caligraphic omicron in lines 162-164 -> no matter how many times I read this part I could not understand what is used for or what it means; (c) There are notions that are mentioned first and introduced later, e.g., reward r_L which shows up in line 145 and later in line 204 is properly explained in Section 2.4 for the first time. In general, I believe that this paper suffers from serious notational issues which can bring confusion to the reader, e.g., I understand what is the difference between states s_S and s_L but it was never made clear what plain s (Equations 1, 2, 3) stands for, to my understanding is the same as s_L but I am also not sure.

**Framework complexity:** Regarding the Communication POMDP framework, I find it overly complicated and poorly motivated, why is this the correct framework to use? Given that the main purpose of this paper is to be the starting point for research on formalizing deception, I believe that a more comprehensive framework would be quite more helpful. Furthermore, the regret notion in Section 2.3 seems quite simple, which does not justify why all this complexity in the proposed framework. Even though, I have not worked out the details, it seems that turn-based Dec-POMDPs [1, 2, 3] with two agents, Listener and Speaker, could be a potentially suitable framework for expressing your regret notion.

**Definition:** I appreciate the honesty in the limitations section, and I think that since you present your solution as a starting point this part is fine. I believe however that your definition has one additional important weakness that is not mentioned in the Limitations section. In case, Speaker tries to deceive Listener, but the latter does not trust the former and hence it is not influenced by its actions, your approach would classify Speaker as not deceitful, even though it is.

[1] Sidford, Aaron, et al. "Solving discounted stochastic two-player games with near-optimal time and sample complexity." International Conference on Artificial Intelligence and Statistics. PMLR, 2020.

[2] Jia, Zeyu, Lin F. Yang, and Mengdi Wang. "Feature-based q-learning for two-player stochastic games." arXiv preprint arXiv:1906.00423 (2019).

[3] Frans A. Oliehoek and Christopher Amato. 2016. A concise introduction to decentralized POMDPs. Springer

**Questions:**

**Q1:** In lines 142 143 is it \pi or \hat{\pi}?

**Q2:** In section 2.4 I do not understand how the reward in Eq. 3, i.e., the belief state, is equivalent to measuring the accuracy of the belief state. Could you maybe further explain this part?

**Q3:** How would you need to adapt your definition in the presence of multiple Speakers, in order to measure the individual degrees of deception?

**Q4:** Why is Communication POMDP the suitable framework for this problem? What is the rationale behind modeling Listener as part of the environment and not as a second acting agent with its own policy and objective?

**Suggestion:** I think it would be useful for the exposition of the experimental results if you bring Figure 3 to the main paper. If I am not mistaken the whole point of ICLR allowing an extra page this year was to account for such things.

---

### Official Review · Reviewer_cidJ · 2024-11-02

**Soundness:** 3
**Presentation:** 3
**Contribution:** 2
**Rating:** 5
**Confidence:** 3

**Summary:**

The paper introduces a formal definition of deception based on the interaction between a speaker and a listener, both modeled as Partially Observable Markov Decision Processes (POMDPs). In particular, a "regret" theory is proposed to quantify deception by measuring the speaker's influence on the listener's beliefs, actions, and utility. Experiments and evaluations were conducted to assess whether the formal "regret" theory aligns with human judgments about deception.

**Strengths:**

Originality:
- The paper introduces a novel approach to quantifying deception within the framework of POMDPs, capturing different forms of deceptive behavior.

Quality:
- Several well-designed experiments were conducted to assess how well the formal definition aligns with human intuition regarding deception.

Significance:
- The work is highly relevant to the fields of AI ethics and safety, providing a foundation for future work.

**Weaknesses:**

- A realistic scenario of deception typically involves multi-step interactions between the speaker and listener. Although the paper states, "While we consider this single-step formulation for simplicity of exposition, it is straightforward to extend the formalism into a sequential setting," it is not clear how the current single-step communication (PO)MDP model could be adapted to capture multi-step interactions. The extension to a sequential setting is not sufficiently demonstrated or explained.

- The paper overlooks the importance of the speaker’s beliefs and intentions. For example, does the speaker believe their statement is false? This omission can lead to misclassifications, such as confusing incompetence with deception, as noted in the paper’s limitations. Furthermore, intent is crucial in legal and ethical definitions of deception, where AI agents may exhibit intentional deception towards users. The paper’s "regret" theory is purely consequentialist, focusing solely on the outcome or utility without considering the speaker's intent. [1][2]

- The experiments involving large language models (LLMs) don't  explore the diversity of deceptive capabilities these models might show. For instance, LLMs can engage in strategic deception under specific conditions, such as when pressured. This aspect needs further  evaluation of LLM deception. [3]

1.Francis Rhys Ward, Francesco Belardinelli, Francesca Toni, and Tom Everitt. Honesty is the best
policy: Defining and mitigating ai deception, 2023.

2. Jaume Masip, Eugenio Garrido, and Carmen Herrero. Defining deception. Anales de Psicología, 2004.
ISSN 0212-9728. URL https://www.redalyc.org/articulo.oa?id=16720112.

3. Large Language Models can Strategically Deceive their Users when Put Under Pressure
https://openreview.net/forum?id=HduMpot9sJ

**Questions:**

1. In Figure 3 of the supplement, you state, "The figure on the left shows conversations from *Deal or No Deal*." Did you mean the figure on the right?

2. Figure 5 in the supplement references "posterior information." Where is the Bayesian model described?

3. How is the reward calculated in the examples?

---

### Official Review · Reviewer_5KSK · 2024-11-02

**Soundness:** 3
**Presentation:** 3
**Contribution:** 2
**Rating:** 5
**Confidence:** 4

**Summary:**

This paper presents a formalisation of deception in terms of POMDPs (partially observable Markov Decision Processes). Two agents, a speaker and a listener have their own PODMDPs. The listener has a model of the speaker's policies, and the speaker has a probability distribution over this model.

The speaker communicates information to the listener. The speaker may be honest or deceptive (including deception by omission), and the listener takes actions based on their model of the speaker's honesty or deception.

Then deception is defined in terms of the listener's regret. This regret is measured, in the paper, via two reward functions: the listener's true reward function (which measures how much they have been harmed by the speaker's possible deception) or a reward function based on the accuracy of the listener's beliefs (which measures how much they have been mislead by the speaker's possible deception).

Some human feedback experiments on simulated data show that human judgments of deception are somewhat correlated with these two regret measures, with the correlation being stronger with the accuracy-of-belief based regret.

**Strengths:**

The formalism is fine and intuitively plausible (though somewhat idealised). The experiments are well executed and well presented.

**Weaknesses:**

The formalism is interesting, but is only a mild variation of multi-agent POMDP (and MDP) formalisms from other papers (see, eg CIRL papers such as "Cooperative Inverse Reinforcement Learning"; searching for competitive MDP and POMDP papers will give many other examples).

It is not that the formalism is exactly the same as previous formalisms, but that it is very similar to many of them. Nevertheless, the formalism is fine (as mentioned above), and, if it lead to powerful examples and demonstrations, would be an excellent introduction to a great paper. But without those powerful examples or demonstrations, it is not enough to make the paper worthwhile in itself. The experiments show that there is a certain overlap between human judgements and these measures (especially the second regret measure with accuracy of beliefs as the reward), but all the correlations, bar one, are below 0.5, and "humans weakly agree with this regret measure in three examples" is not enough meat on the bones for this paper.

What the paper needs is a great use case for this formalism, powerful experiments that show its use. An intuitively plausible definition is not enough.

**Questions:**

Find a more powerful use case for the formalism. Fine-tuning an LLM with it, running it over a very large database of deceptions, showing its validity when compared with other literatures (eg the psychological literature), running it on online games where there is a way for players to demonstrate trust or lack thereof,... These are some suggestions, and I'm sure that the authors can come up with better ones. Find results so powerful that readers don't think "this is a plausible and interesting formalism", but "wow, this formalism really illuminates things/leads to impressive improvements".

---

### Official Review · Reviewer_fzYT · 2024-11-04

**Soundness:** 3
**Presentation:** 3
**Contribution:** 2
**Rating:** 3
**Confidence:** 4

**Summary:**

This paper presents a new quantitative definition of deception in agent-to-agent dialogue interactions modelled as partially observable Markov decision processes (POMDPs). One of the agents is the speaker (S), and the other agent is the listener (L). The listener L is modelled as a POMDP over a certain set of world states. The set of observations of the POMDP representing L is the same as the set of actions available to S. The speaker S is modelled as a POMDP over a state space that contains the world states plus belief states over L. Essentially, S has access to the ground truth of the world and the actions provided by L, and has a belief model over the beliefs and policies of L, while L has no access to the ground truth, and has a belief model over the ground truth guided by the actions of S.

Each agent (S and L) has its own reward function.

Degree of deception is defined as the difference between the expected reward of L if it listens to (and therefore updates its beliefs over) S, and the expected reward of L if it does not listen to (and therefore does not update its beliefs over) S. This way, a positive value indicates an altruistic speaker (S makes the reward of L larger if L listens to S), a negative value indicates a deceptive speaker (S makes the reward of L smaller than it would be if L did not listen to S), or neutral if they are the same. (*)

The paper states that different concepts of deceptiveness can be captured by plugging different reward functions in Eq. (1). In particular, deceptiveness can be defined as (i) S producing worse outcomes (reward for deception = reward for the task), (ii) S producing beliefs in L further from the truth (reward for deception = belief is close to reality), or (iii) a combination of both (i) and (ii).

Armed with this definition, the paper presents an experimental study, where they generate scenarios with different pairs of agents, ask human subjects to rate the degree of deceptiveness and compare the results with the degree of deceptiveness given by Eq. (1), as well as the degree of deceptiveness when substituting human subjects by state of the art large language models (LLMs). The paper claims that their results support the hypothesis that their definitions of deceptiveness align with human intuition, especially when the reward combines outcome and beliefs, and that the alignment with human intuition is much better than that given by state of the art LLMs.

-------
(*) This is inverted to what is stated in the text (lines 198-209). I believe there is either a negative sign missing in Eq. (1) or a mismatch in describing the equation in lines 208-209. In any case, it is at the level of a typo, it does not significantly affect the contribution or quality of the paper.

**Strengths:**

S1. The topic of deception in autonomous systems is relevant and timely. The use of autonomous systems in day-to-day decision-making is increasing (especially with the current development of LLMs), and it is fundamental to have models of different intentional harms to increase trust in these systems by the public and accountability for potential harms caused by them. Deceptive language, especially in interactive systems, is of particular relevance.

S2. The paper is written nicely, with a strong motivation, clear structure and understandable examples to guide the reader.

S3. As an experimental evaluation, the paper includes a study on human subjects and addresses state of the art LLMs. The scenarios presented are a good balance of simple and realistic.

S4. The paper has a fair discussion on the limitations of their approach, mentioning how there is work to be done to deploy the proposed concept in more complex and realistic scenarios.

S5. The paper engages with current literature on different definitions of deception and use of LLMs in relation to deception.

**Weaknesses:**

**MAIN WEAKNESSES**

W1. The formalism has no clear novelty. Defining deceptiveness as the level of regret just passes the ball of defining deceptiveness to the reward function. This is not in itself a bad decision, but it does mean that the interest does not lie so much in the definition as proposed in Eq. (1), but rather lies in the choice of reward function. The choice of reward function feels underexplored to me as part of the experimental report.

W2. I think the experimental evaluation goes in the right direction, and most of the data obtained will be useful, but I find it weak as it is now. I will structure my criticism into points that I believe are misleading and points that I believe are incomplete. I also number them, to facilitate later discussion.


W2.1. Misleading.

- W2.1.1. In Table 1, the numbers presented indicate the correlation between human perceived deceptiveness and the values given by regret and by LLMs. In the table it states that these results were statistically significant, with a p-value < 0.001. I do not see in the text what statistical test this p-value refers to, I can only assume by context that the null hypothesis was "there is no correlation between human ratings and machine ratings". If this is the case, the result is hardly surprising (although it is useful as a means of a sanity check), and I find it misleading to accompany it to the concrete values given in this table.

- W2.1.2. The nutritionist scenario is a bad choice, and I do not agree with the reason given to its lower correlation in lines 361-363. In the nutritionist scenario, human subjects are presented with facts that are controversial in the current public opinion (whether protein, restriction of carbohydrates or herbal teas boost energy). The human subject is going to come with its own beliefs to the task, and they would certainly influence their perception of deceptiveness. I think the nutritionist example is a bad choice and without having prior information on the beliefs of the human subjects with respect to protein, carbohydrates and teas, no reliable conclusions can be extracted from the data of that experiment.

- W2.1.3. Lines 355-356 state that "We largely find that a combined regret formulation better captures human intuitive notions of deception across all three scenarios, confirming our hypothesis from Section 2.3 that both belief and task reward contribute to improving the correlation with human judgment". While it is true that the "Combined" column is larger than the "Belief" column, it is not by much. It would be helpful to accompany this statement with a statistical test and its corresponding p-value.


W2.2. Incomplete.

- W2.2.1. Lines 368-373 include information about multi-step conversations. It would be useful to have a table similar to Table 1 summarizing the information, maybe in an appendix if it does not fit in the main text. Also, a conclusion is given that the correlation between humans and regret is higher for multi-step conversations. It would be helpful to accompany this statement with its corresponding statistical test and p-value. It would also be interesting to know how much (if any) the LLMs improve in multi-step conversation.

- W2.2.2. One of the conclusions derived from the experimental evaluation is that the presented regret-based formalism aligns better with human intuition than the estimation given by LLMs. Again, it would be interesting to know the statistical significance of this statement, but more importantly, it would be interesting to understand why. Given the 1-5 scale, it is possible that the LLM produces a less extreme (but still on the correct side) value than the human (for example, the LLM would choose 2 instead of 1, or 4 instead of 5). This would produce a smaller correlation, while indicating that the LLMs are still aligned with human intuition. Another possibility is that the lower correlation comes from the LLMs contradicting human intuition (i.e. the LLM choosing a value >3, when the human chooses a value <3, and vice versa). Of course, in reality, it is probably a combination of both phenomena. It would be however very informative to include some information about this, maybe as part of a qualitative analysis (currently Sec. F in the appendix).

- W2.2.3. One of the questions in the study is which reward function produces deceptiveness degrees that best align with human intuition, and the winner is the "combined one". However, as far as I can tell, it is not stated in the paper what is the weight used in combining these values. As an extra step, it would also be interesting to see how the correlation varies for different weights, and whether an "optimal" weight arises from the experiments.


I hope the authors do not get discouraged by this review. I really like the study presented in the paper and think it has much value.  With some additional depth and clarity in the experimental evaluation, this paper could be a strong contribution for a top-tier venue like ICLR in the future.


**OTHER (MINOR) REMARKS**

These are smaller remarks, mostly editing issues. I hope the feedback serves to polish the paper.

R1. The bibliography needs to be polished. Here is a list of issues I found.

- R1.1. There are repeated items, for example [He et al. 2018], [Lewis et al. 2017], [Wang et al. 2020].
- R1.2. There are many items that lack a journal, conference, arxiv id or similar. I know in the era of the internet one can find papers just from the title, but let's keep good practices. For example [Abdulhai et al. 2023], [Amodei et al. 2016], [Bai et al. 2022], [Sung et al. 2023], [Pan et al. 2023], [Park et al. 2023], [Touvron et al. 2023], [Ward et al. 2023], [Wei et al. 2023].
- R1.3. There are typos: <i>diplomacy</i> in [Bakhtin et al. 2022], missing capitalization in [Greene 2007], [Wang et al. 2021] .
- R1.4. This may be a quirk of mine, but I find it misleading to cite papers that have been presented at major AI/ML venues giving only their arxiv id, while other papers are cited in the proceedings of some conference or journal. For example [Aakanksha et al. 2022] appeared in JAIR 2024, [He et al. 2018] in EMNLP 20218, [Lin et al. 2021] in ACL 2022, or [Pan et al. 2023] in ICML 2023. I would appreciate at least consistency.
- R1.5. I don't think [Brown et al. 2020] is an appropriate citation for GPT3.5-Turbo. See the discussion here for example: https://community.openai.com/t/how-to-cite-text-davinci-003-in-academic-paper/369821.


R2. Apart from the missing details mentioned in W2, there are some missing definitions. When defining POMDP (line 104), the set of beliefs (B) is mentioned, without stating what it is. I assume that a belief is a probability distribution over states (as usual), but this should be stated for the sake of completeness. On a similar note, it is not clear to me what b_L(s) means in Eq.(3). To my understanding, b_L is a probability distribution of states, and b_L(s) is the probability of state s. If so, I would find it more suitable for the reward function to be a distance between the probability distribution b_L and the probability distribution where "s" has probability 1, and the rest of the states have probability zero. This is a small change, but as a reader, I spent some time having to think over the definition because it was missing. This distracting confusion could be easily avoided by providing the complete definitions.

R3. The authors could consider engaging with the recent literature of explainable RL in terms of deception and intentional behaviour, and how this can be used to analyse harmful behaviours (deception being here a harmful behaviour). See for example:
- Liu, Z. et al. Deceptive Reinforcement Learning for Privacy Preserving Planning. AAMAS 2021.
- Lewis, A. et al. Deceptive Reinforcement Learning in Model-Free Domains. ICAPS 2023.
- Cordoba, F.C. et al. Analyzing Intentional Behavior in Autonomous Agents under Uncertainty. IJCAI 2023.
- Beckers, S. et al. Quantifying Harm. IJCAI 2023.


R4. These are just two suggestions:
- I would make section 2.4 significantly shorter to allow for more details on the experimental evaluation. The definition is easy enough to grasp, it may not need so much redundant explanations.
- In lines 256-257 I would put a different example.

R5. The paper needs an editorial pass. Here is a list of typos and the like.
- obtains --> obtained (line 079)
- Figures 3, 4, and 5 are mentioned in the main text without stating that they are part of the appendix.
- Reference to Fig. 4 in line 315-316 should be to Fig. 6.
- I do not understand the point of Fig. 4 at all. Does it add any insight that is not already provided by Fig. 6?
- left --> right (line 931).
- The caption of Fig.4 is misleading, and seems to be referring to Fig. 5 instead.
- In Figure 7, the image representation of practising photography and being part of community events are swapped. If this is not only a typo in the paper, but this is also how the examples were presented to the human subjects and LLMs, this fact should be stated somewhere, or the experiment repeated.
- Table 2 is not mentioned in the text.
- The font in Figures 3 and 4 is significantly smaller than the main text. Consider making it larger to improve readability. Especially in the appendix, where the page limit does not apply.

**Questions:**

Q1. How did you choose the reward functions in the combined metric? Was any hyperparameter tuning done for the relative weight of the outcome and belief terms of the combined reward?

Q2. Can you provide some insight in why the correlation with LLMs is so low?

Q3. Was there any demographic selection criteria for the participants in the study, like concrete age ranges, gender, race, etc?

Q4. Any comment about the missing information in W2 is welcomed.

---

### Meta-Review · Area_Chair_AQjY · 2024-12-21

**Metareview:**

The reviewers of this paper provided detailed and high-quality feedback. The main concerns raised are as follows:

- The formalization is relatively straightforward and lacks significant novelty.
- The experimental evaluations include several misleading or incomplete aspects (see Reviewer fzYT’s comments).
- The presentation could be significantly improved and simplified. For instance, while the formulation is based on POMDP, the world state remains static in both the theoretical framework and the experiments. If the state does not transition, why base the formulation on POMDP in the first place?

The authors didn't respond to any above concerns, so it's a clear reject.

**Additional Comments On Reviewer Discussion:**

The paper's formalization is relatively straightforward and lacks significant novelty. The experimental evaluations contain several misleading or incomplete aspects, as highlighted by Reviewer fzYT. Additionally, the presentation could be significantly improved and simplified. For example, while the formulation is based on POMDP, the world state remains static in both the theoretical framework and the experiments, raising the question of why a POMDP-based formulation is used if the state does not transition.

The authors provided no response to the above criticism.

---

### Decision · Program_Chairs · 2025-01-22

Reject